# Mitigating Gradient Interference for Efficient Sparse Fine-Tuning of Large Language Models

## Abstract

Large Language Model (LLM) sparsification plays a crucial role in model compression. Among various methods, training-free approaches are highly efficient but often result in accuracy loss, while full fine-tuning requires substantial computational resources. Recent works have begun exploring sparse Parameter-Efficient Fine-Tuning (PEFT) methods, but lack theoretical guidance. This study presents the first comprehensive theoretical framework for efficient sparse fine-tuning, addressing a critical gap in the literature. Specifically, we identify gradient conflict as the primary issue in PEFT sparse methods, wherein masked pretrained weights and corresponding PEFT weights exhibit competing optimization objectives during fine-tuning, potentially compromising model performance. We theoretically model this phenomenon and identify three key factors influencing the efficacy of fine-tuning in sparsified LLMs: (1) error introduced by weight norms, (2) error composition from PEFT structures, and (3) error accumulation during fine-tuning. Leveraging these theoretical insights, we propose a novel iterative sparse fine-tuning scheme that systematically addresses each identified factor. We implement an iterative process alternating between sparsity and fine-tuning to mitigate accumulated error in single turn of finetuning. We employ pooling instead of low-rank decomposition to reduce error composition from PEFT structures. We apply normalization to PEFT modules during fine-tuning, constraining error values by limiting weight norms while preserving representational capacity. Additionally, we utilize Centered Kernel Alignment based information similarity assessment for adaptive allocation of layer-level sparsity and PEFT parameter quantities, addressing layer-specific redundancy. Empirical evaluation on a 50% sparse LLaMA-2 7B model demonstrates the superiority of our approach, achieving lossless compression.

## 1 Introduction

The field of deep learning has witnessed an unprecedented surge in model sizes (Zhang et al., 2022; Chiang et al., 2023; Touvron et al., 2023b; Achiam et al., 2023) , leading to significant advancements in various domains(Ma et al., 2023; Xia et al., 2023; Zhang et al., 2023a; Guo et al., 2023; Frantar & Alistarh, 2023; Sun et al., 2023; Zhang et al., 2023c). However, this growth has also introduced substantial challenges in terms of storage requirements and computational demands. As models continue to expand, the need for efficient optimization techniques has become increasingly critical. Sparsification techniques (Frantar & Alistarh, 2023; Sun et al., 2023; Ma et al., 2023; Xia et al., 2023; Zhang et al., 2023c) have shown promise in reducing model size and computational complexity by eliminating redundant or less important parameters. Concurrently, PEFT methods (Houlsby et al., 2019; Lester et al., 2021; Hu et al., 2021) have emerged as effective strategies for adapting pre-trained models to specific tasks with minimal parameter updates. The integration of these approaches presents a compelling opportunity to simultaneously achieve model compression and task adaptation, potentially revolutionizing the deployment of large-scale deep learning models across diverse applications.

The direct application of iterative sparsification followed by parameter-efficient module fine-tuning faces significant limitations, primarily due to the emergence of the "Sparse Weight Gradient Interfer-

ence" phenomenon. This issue manifests when $\Delta W$ exhibits gradients corresponding to zero-valued pre-trained weights, leading to interference with gradients at other positions. The root cause of this phenomenon lies in the nature of the fine-tuning process. While pre-trained model weights are frozen and sparsified, subsequent fine-tuning of $\Delta W$ continues to generate gradients in areas where original weights were set to zero. This results in PEFT module parameters corresponding to sparse zero-weight positions computing gradients, causing interference between these gradients, affecting optimization direction, and increasing loss error.

To rigorously understand this phenomenon, we conducted a theoretical analysis of the impact of sparse $\Delta W$ on model fine-tuning loss, deriving loss error bounds. Our analysis suggests three primary directions for error reduction: (1) Implementation of iterative sparsification and fine-tuning to avoid introducing excessive errors at once, (2) Structural improvements on PEFT modules to reduce the interference impact on other parameters, and (3) Introduction of regularization constraints on PEFT module parameters. These insights provide a foundation for developing more effective sparse fine-tuning methods for large language models.

Based on our observations and theoretical analysis, we propose an efficient sparse adaptation method designed to mitigate gradient conflict issues. Our method addresses the limitations of existing methods by incorporating several key innovations. At the core of our method is a parameterizable small square weight matrix that serves as the PEFT module, operating in linear projection, applying simple pooling operations to the input. This process involves linear mapping of pooled inputs followed by inverse pooling, effectively reducing the impact of gradients from sparsified positions on other PEFT model parameter positions. We implement an iterative sparsification and fine-tuning process, gradually increasing sparsity rates. This approach minimizes the total error introduced in each fine-tuning round. Consistent with previous work, we employ a cubic schedule for increasing sparsity ratios. Following the final sparsification round, we apply extended fine-tuning steps to fully leverage the fine-tuning process and enhance model capabilities. During the fine-tuning phase, we apply regularization to weights. This process constrains the parameter matrix norm, thereby reducing gradient errors. The combination of these techniques allows our method to effectively address the "Sparse Weight Gradient Interference" phenomenon while maintaining model performance. This work contributes in three key ways:

- We provide an in-depth investigation of post-sparsification fine-tuning issues, identifying and analyzing the "Sparse Weight Gradient Interference" phenomenon. Our theoretical analysis offers both loose and tight bounds on its impact on model fine-tuning loss, providing valuable insights for future improvements.

- We propose a novel method combining iterative sparsification, pooling based PEFT module, and regularization, which effectively addresses gradient conflict issues and utilizes CKA metrics for adaptive computation of MoRA rank and layer-wise sparsity.

- Our experimental results demonstrate our method's significant improvements over baseline methods in model effectiveness, computational efficiency, and parameter utilization rate. These advancements offer a comprehensive solution to the challenges of model compression and task adaptation in deep learning.

## 2 MODELING THE SPARSE FINE-TUNING PROCESS

Building upon the framework introduced by former work like SparseGPT (Frantar & Alistarh, 2023), we formulate the problem as a layer-wise reconstruction task, aiming to minimize the discrepancy between sparse and dense LLM layers. Consider an LLM with $L$ layers, where $\mathbf{W}_i \in \mathbb{R}^{C_{\text{out}} \times C_{\text{in}}}$ denotes the weight matrix of the $i$-th layer, and $\mathbf{X}_i \in \mathbb{R}^{C_{\text{in}} \times D}$ represents the input feature maps. $C_{\text{in}}$, $C_{\text{out}}$, and $D$ correspond to the number of input channels, output channels, and hidden dimension, respectively. Sparsity is introduced through a binary mask $\mathbf{M}_i \in \{0, 1\}^{C_{\text{out}} \times C_{\text{in}}}$ applied to $\mathbf{W}_i$. We extend this framework by incorporating paradigm of PEFT ($\Delta\mathbf{W}_i$). Let $\mathbf{P} = (p_1, \ldots, p_L)$ represent the ranks across all layers, and $\mathbf{S} = (s_1, \ldots, s_L)$ denote the corresponding sparsity rates. The PEFT adaptation problem can then be formulated as:

$$\min_{\mathbf{M}, \mathbf{W}} \sum_{i=1}^{L} \|\mathbf{W}_i * \mathbf{X}_i - (\mathbf{M}_i \odot (\mathbf{W}_i + \Delta\mathbf{W}_i)) * \mathbf{X}_i\|_2^2 \,, \, s.t. \, 1 - \frac{\|\mathbf{M}_i\|_0}{C_{\text{out}} \cdot C_{\text{in}}} = s_i, T(\mathbf{S}) = \Theta, T(\mathbf{P}) = \Omega.$$

$$(1)$$

where, $*$ denotes matrix multiplication, $\odot$ represents element-wise multiplication, and $\|\cdot\|_2$ signifies the $\ell_2$ norm. $T(\mathbf{S}) = \Theta$ constrains the average sparsity rate across layers to $\Theta$, while $T(\mathbf{P}) = \Omega$ limits the parameter budget during fine-tuning to $\Omega$. The optimization process requires determining three key parameters: the sparsity mask $\mathbf{M}$, how $\mathbf{\Delta W}_i$ is composed, and layer-wise budgets allocations $\mathbf{R}, \mathbf{S}$.

To effectively solve the joint optimization problem, we reformulate it as a bi-level optimization task. This approach allows us to leverage existing techniques for sparsity mask optimization while incorporating PEFT weight optimization. The bi-level optimization is structured as followed:

**Upper-level.** Sparsity mask optimization $\mathbf{M}_i$ for each layer $i$, which formulated as:

$$\min_{\mathbf{M}} \sum_{i=1}^{L} \|\mathbf{W}_i * \mathbf{X}_i - (\mathbf{M}_i \odot \mathbf{W}_i) * \mathbf{X}_i\|_2^2, \ s.t. \ 1 - \frac{\|\mathbf{M}_i\|_0}{C_{\text{out}} \cdot C_{\text{in}}} = s_i, T(\mathbf{S}) = \Theta. \tag{2}$$

**Lower-level.** PEFT weight $\mathbf{\Delta W}$ optimization using calibration data to minimize the Next Token Prediction Loss, which is formulated as:

$$\min_{\mathbf{\Delta W}} \mathcal{L}_{\text{NTP}}(\mathbf{M} \odot \mathbf{W} + \mathbf{\Delta W}, \mathcal{D}) , \ s.t. \ T(\mathbf{P}) = \Omega. \tag{3}$$

where $\mathcal{L}_{\text{NTP}}$ represents the Next Token Prediction Loss, $\mathcal{D}$ is the calibration dataset, and the constraint $T(\mathbf{P}) = \Omega$ ensures the parameter budget during fine-tuning is limited to $\Omega$.

The intuitively optimal solution is naive integration of the sparsity mask directly with PEFT weights through dot production ($\mathbf{M} \odot \mathbf{\Delta W}$). However, this solution introduces critical challenges that undermine the core benefits of Parameter-Efficient Fine-tuning:

1. **Dimensional Expansion:** This approach forces $\mathbf{\Delta W}$ to match the dimensionality of the original weight matrix $\mathbf{W}$, negating the compactness advantage of PEFT methods.

2. **Memory Inefficiency:** The expansion of $\mathbf{\Delta W}$ significantly increases memory requirements, rendering the approach impractical for large-scale LLMs on standard hardware.

3. **Computational Overhead:** Element-wise multiplication between large matrices introduces additional computational steps, degrading performance and increasing latency.

The proposed low-level approach that do not impose mask to $\mathbf{\Delta W}$ , while avoiding above pitfalls, still presents a significant challenge in the context of sparsed pre-trained weights and dense PEFT weights. This challenge is a phenomenon termed "Sparse Weight Gradient Interference" (SWGI).

SWGI arises from the mismatch between the sparse structure of the pre-trained weights and the dense nature of the PEFT module weights. In the current formulation, the PEFT module weights ($\mathbf{\Delta W}$) are applied uniformly across all positions, including those where the pre-trained weights are masked out (i.e., set to zero) by the sparsity mask $\mathbf{M}$. This incongruence leads to suboptimal utilization of the fine-tuning capacity and potentially introduces noise into the model.

Specifically, SWGI occurs when PEFT module weights at positions corresponding to masked pre-trained weights continue to calculate and propagate gradients during the fine-tuning process. To address SWGI and its implications, we conduct a comprehensive theoretical analysis in the following section. This analysis aims to provide a deeper understanding of the phenomenon and lay the groundwork for more effective solutions.

# 3 ANALYSIS FOR SPARSE WEIGHT GRADIENT INTERFERENCE

In this section, we analyze the phenomenon named "Sparse Weight Gradient Interference" in theoretical and experimental ways. We firstly give the preliminaries and basic format of loss errors for general reparameterizable PEFT methods, summarize the key factor(1) "error introduced by weight norms". Then we analysis the error of LoRA(Hu et al., 2021) and our method, come to the key factor(2) "error composition from PEFT structures". Finally, we prove the key factor(3) "error accumulation during fine-tuning".

### 3.1 PRELIMINARIES

For simplicity, let $W \in \mathbb{R}^{D \times D}$ be the original weight matrix of a LLM. The sparsification mask is $M \in \{0,1\}^{D \times D}$, where elements set to 0 indicate sparse positions. Without considering effects in actual implementation, the optimal loss in the ideal case is:

$$\mathcal{L}_{\text{ideal}} = \mathcal{L}(\mathbf{M} \odot (\mathbf{W} + \mathbf{\Delta W})) \tag{4}$$

We aim to analyze the discrepancy between the loss computed using $\mathbf{M} \odot \mathbf{W} + \mathbf{\Delta W}$ and the ideal loss that would be obtained in $\mathbf{M} \odot (\mathbf{W} + \mathbf{\Delta W})$ where $\Delta W$ were zero at sparse positions. We denote the set of sparse positions as $Z$, using it to indicate operations or values specific to these positions. Furthermore, we denote $\mathbf{M} \odot (\mathbf{W} + \mathbf{\Delta W})$ as $\mathbf{W}_{ideal}$. The loss function we actually optimize and its Taylor expansion can be expressed as:

$$\mathcal{L}_{\text{actual}} = \mathcal{L}(\mathbf{W}_{ideal} + \mathbf{\Delta W}_Z)$$

$$\approx L(W_{ideal}) + \nabla_{W_Z} L(W_{ideal}) \cdot \Delta W_Z + \frac{1}{2} \Delta W_Z^\top H_Z \Delta W_Z \tag{5}$$

where $\nabla_{W_Z} L(W_{ideal})$ represents the gradient of the loss function at positions $Z$, and $\Delta W_Z$ denotes weights of $\Delta W$ at these positions.

Based on the loss function presented in Equation 5, we can proceed to analyze three key factors that significantly influence the manifestation and impact of Sparse Weight Gradient Interference. These factors, which will be examined in detail in the subsequent sections, provide a comprehensive framework for understanding the complex interactions between sparse pre-trained weights and dense PEFT modules.

### 3.2 UPPER BOUND OF ERROR INTRODUCED BY $\Delta W$ NORMS

**Theorem 3.1** (Loss Error Bound). *The loss error of $\Delta L$ introduced by SWGI is bounded as follows:*

$$|\Delta L| \leq ||\nabla_{\mathbf{W}_Z} \mathcal{L}(\mathbf{W}_{ideal})|| \cdot ||\mathbf{\Delta W}_Z|| + \frac{1}{2} ||\mathbf{H}_Z|| \cdot ||\mathbf{\Delta W}_Z||^2 \tag{6}$$

*where: $H_Z$ is the Hessian matrix restricted to the sparse positions $Z$, $|| \cdot ||$ represents an appropriate matrix or vector norm (e.g., Euclidean norm for vectors and spectral norm for matrices).*

*Proof.* For the specific proof process, please refer to Appendix.A.

This theorem provides a theoretical foundation for understanding the error introduced by weight norms in PEFT sparse methods, specifically addressing factor (1) from our identified key factors. The analysis reveals:

1. **Direct Relationship with Weight Norms:** Theorem3.1 shows that the error bound is directly related to $||\Delta W_Z||$, the norm of the weight adjustments. This demonstrates that the magnitude of weight changes directly influences the potential error.

2. **Weight Gradient Influence:** The term $||\nabla_{W_Z} L(W_{ideal})||$ in bound indicates that the error is also dependent on the gradient magnitude at the sparse positions. This suggests that areas of the loss landscape with steeper gradients are more susceptible to larger errors.

3. **Trade-off Between Adjustment and Error:** These bounds illustrate the fundamental trade-off in PEFT methods: larger weight adjustments may allow for more significant model changes but at the cost of potentially larger errors.

Furthermore, Theorem3.1 also point to the potential influence of the specific structure of the PEFT module that we will discuss in next subsetion.

### 3.3 ERROR BOUND FROM STRUCTURE OF LORA AND OUR METHOD

In this subsection, we firstly give the loss error bound for LoRA, then we give bound for our method, lastly we compare these bounds and prove that our method has more strict bound than LoRA, showing the superior of our method.

### 3.3.1 ERROR BOUND FROM STRUCTURE OF LoRA

LoRA modifies the weight matrix $W$ by introducing a low-rank update $\Delta W$, which is decomposed into the product of two low-rank matrices $A$ and $B$: $\Delta W = AB$, where:$A \in \mathbb{R}^{D \times r}$ and $B \in \mathbb{R}^{r \times D}$ are low-rank matrices with rank $r \ll D$. The adjustment $\Delta W$ is thus a rank-$r$ matrix, enabling efficient storage and computation. We start with first-order approximation in case of simplicity.

**Lemma 3.2** (Loss Error Bound for LoRA). *The absolute value of the loss error $\Delta L$ introduced by the LoRA adjustment is bounded as follows:*

$$|\Delta L| \leq \|\nabla_{W_Z} L(W_{ideal})\| \cdot \|M \odot (AB)\| \tag{7}$$

Building upon this lemma, we can further refine the error bound by analyzing the norms of $A$ and $B$, and the sparsity in $M$.

**Theorem 3.3** (Refined Loss Error Bound). *The absolute loss error introduced by the LoRA adjustment in a sparse neural network is bounded as follows:*

$$|\Delta L| \leq \|\nabla_{W_Z} L(W_{ideal})\| \cdot \sqrt{k} \|A\|_2 \|B\|_2 \tag{8}$$

*where $k$ is the number of non-zero elements in $M$, $\|\cdot\|_2$ denotes the spectral norm.*

*Proof.* For the specific proof process, please refer to Appendix.B.

### 3.3.2 ERROR BOUND ANALYSIS FOR OUR METHOD

We present a PEFT method that operates on input vector $\mathbf{X} \in \mathbb{R}^D$ with three main steps:

1. **Pooling Operation**: We partition $\mathbf{X}$ into $\mathbf{g}$ equal-sized blocks, each of size $n$, such that $D = \mathbf{g} \times n$. The pooling operation computes:

$$\mathbf{X}_1 = \frac{1}{n} \mathbf{X}_{\text{reshaped}}^\top \mathbf{1}_n \in \mathbb{R}^{\mathbf{g}} \tag{9}$$

where $\mathbf{X}_{\text{reshaped}} \in \mathbb{R}^{n \times m}$ and $\mathbf{1}_n \in \mathbb{R}^n$ is a vector of ones. Note: To facilitate our analysis, we only consider the case where D is divisible by g.

2. **Linear Transformation**: We transform the pooled vector using weight $\mathbf{G} \in \mathbb{R}^{\mathbf{g} \times \mathbf{g}}$:

$$\mathbf{Y}_1 = \mathbf{G} \cdot \mathbf{X}_1 \in \mathbb{R}^{\mathbf{g}} \tag{10}$$

3. **Expansion Operation**: We expand $\mathbf{Y}_1$ back to the original dimension $D$:

$$\mathbf{Y} = \text{Expand}(\mathbf{Y}_1) = \mathbf{Y}_1 \otimes \mathbf{1}_n \in \mathbb{R}^D \tag{11}$$

where $\otimes$ denotes the Kronecker product.

Thus, the corresponding adjustment matrix $\Delta \mathbf{W} \in \mathbb{R}^{D \times D}$ is:

$$\Delta \mathbf{W} = \frac{1}{n} \mathbf{M} \odot (\mathbf{G} \otimes \mathbf{1}_n) \in \mathbb{R}^{D \times D} \tag{12}$$

where $\odot$ denotes the Hadamard product, and $\mathbf{1}_n \in \mathbb{R}^{n \times n}$ is a matrix of ones.

We now present a theorem bounding the loss discrepancy $\Delta L$ introduced by our PEFT adjustments.

**Theorem 3.4** (Loss Error Bound from PEFT Structure). *Let $Z$ denote the set of sparse positions in $W$. The absolute value of the loss error $\Delta L$ introduced by our method is bounded by:*

$$|\Delta L| \leq \|\nabla_{W_Z} L(W_{ideal})\| \cdot \sqrt{\frac{k}{n}} \|\mathbf{G}\| \tag{13}$$

*where $k$ is the number of non-zero elements in $M$, and $\mathbf{G}$ is the trainable transformation matrix.*

*Proof.* For the specific proof process, please refer to Appendix.B.

### 3.3.3 ERROR BOUND COMPARISON BETWEEN LoRA AND OUR METHOD

In this section, we provide a rigorous comparison of the loss error bounds derived for Low-Rank Adaptation (LoRA) and our proposed method under constraints of actual implementation. According to our experiment settings, the matrix dimensions and the constraint $r \times D = \mathbf{g} \times \mathbf{g}$. We proceed to relate the norms of the respective adjustment matrices.

Assume that the spectral norms of $A$ and $B$ in LoRA, and $\mathbf{G}$ in our method, are bounded by the same constant $C$:

$$\|A\|_2 \leq C, \quad \|B\|_2 \leq C, \quad \|\mathbf{G}\|_2 \leq C \tag{14}$$

Under this assumption, we can refine the error bounds:

**Corollary 3.4.1** (Refined Bound of LoRA and our method). *Error bounds can be expressed as:*

$$|\Delta L_{LoRA}| \leq \|\nabla_{W_Z} L(W_{ideal})\| \cdot \sqrt{k} \cdot C^2$$
$$|\Delta L_{our}| \leq \|\nabla_{W_Z} L(W_{ideal})\| \cdot \sqrt{\frac{k}{n}} C \tag{15}$$

**Corollary 3.4.2** (Superiority of our method). *For sufficiently large $k$ and a reasonable constant $C$, the loss error bound for our method is significantly smaller than that of LoRA:*

$$|\Delta L_{our}^{Bound}| \leq \frac{1}{\sqrt{n}C} |\Delta L_{LoRA}^{Bound}| \tag{16}$$

This inequality demonstrates that the loss error bound for our method is inversely proportional to $\sqrt{n}$ and scaled by $\frac{1}{C}$ relative to LoRA's bound. Consequently, for sufficiently $n$ and a reasonable constant $C$, $|\Delta L_{\text{our}}|$ is significantly smaller than $|\Delta L_{\text{LoRA}}|$.

*Note:* The above comparison assumes that the spectral norms of the adjustment matrices are bounded by the same constant $C$. In practice, the specific values of these norms may vary, and additional factors such as the choice of hyperparameters could influence the actual error bounds.

### 3.4 ERROR ACCUMULATION DURING FINE-TUNING

In the context of PEFT applied to sparsified Large Language Models (LLMs), error accumulation during fine-tuning poses a significant challenge. Specifically, each fine-tuning step introduces a small adjustment $\Delta W$ at the sparse positions $Z$, which can accumulate over multiple iterations, leading to a substantial deviation from the ideal weight configuration. The following theorem formalizes this phenomenon by providing an upper bound on the accumulated loss error after $T$ fine-tuning steps.

**Theorem 3.5** (Error Accumulation During Fine-Tuning). *During fine-tuning, at each step $t = 1, 2, \ldots, T$, an adjustment $\Delta W_Z^{(t)}$ is applied to the sparse positions, resulting in an adjusted weight:*

$$W^{(t)} = W_{ideal} + \sum_{k=1}^{t} \Delta W_Z^{(k)} \tag{17}$$

*Assume that for each step $t$, the loss function $L$ satisfies the following conditions at $W^{(t-1)}$:*

1. *The gradient $\nabla_{W_Z} L(W_{ideal}^{(t-1)})$ is bounded by $\|\nabla_{W_Z} L(W_{ideal}^{(t-1)})\| \leq G$.*

2. *The adjustment norm is bounded by $\|\Delta W_Z^{(t)}\| \leq \delta$.*

3. *The Hessian $H_Z^{(t)}$ satisfies $\|H_Z^{(t)}\| \leq H_{\max}$.*

*Then, the accumulated loss error $\Delta L_T$ after $T$ fine-tuning steps satisfies:*

$$|\Delta L_T| \leq T \cdot G \cdot \delta + \frac{T(T-1)}{2} \cdot \delta^2 \cdot H_{\max} \tag{18}$$

*Proof.* For the specific proof process, please refer to Appendix.C.

## 4 METHODOLOGY

### 4.1 POOLING-BASED PEFT STRUCTURE

To facilitate efficient fine-tuning of sparsified large language models (LLMs), we introduce a pooling-based Parameter-Efficient Fine-Tuning (PEFT) structure. This approach integrates pooling, linear transformation, and expansion operations into a single adjustment mechanism, thereby reducing computational overhead and mitigating gradient interference. The theoretical underpinnings of this structure are examined in Section 3.3.2.

### 4.2 NORMALIZATION OF PEFT MODULES

Normalization plays a pivotal role in stabilizing the fine-tuning process of Parameter-Efficient Fine-Tuning (PEFT) modules, particularly within sparsified large language models (LLMs). Building upon the theoretical insights discussed in Section 3.2, we employ established normalization techniques like weight-decay or drop-out to limit the magnitude of weight adjustments. This approach mitigates errors introduced by weight norms and helps preserve the model's representational capacity.

### 4.3 ADAPTIVE LAYER-WISE SPARSITY AND PEFT PARAMETER ALLOCATION

In this part, we firstly discuss how to get sparsity rate for each layer via CKA. Next, we introduce adaptive allocation of PEFT parameters based on reconstruction loss from sparsity stage.

#### 4.3.1 INFORMATION THEORY GUIDED SPARSITY RATE SETTING

The generalized Information Bottleneck (IB) (Tishby et al., 2000; Zheng et al., 2021) principle provides a framework for balancing the compression of input representations with the retention of task-relevant information during the sparsification of Large Language Models (LLMs). Let $X$ and $Y$ denote the input and output feature maps of a dense model, while $\tilde{X}$ represents the feature maps of a sparse model. The goal of sparsification is to identify a sparse $\tilde{X}$ that minimizes information redundancy while maintaining the essential relevant information, which can be formalized as follows:

$$\min_{p(X_i|X)} \sum_{i=1}^{L} \sum_{j=i+1}^{L} \left( I\left(\boldsymbol{X}; \tilde{\boldsymbol{X}}_i\right) + I\left(\tilde{\boldsymbol{X}}_j; \tilde{\boldsymbol{X}}_i\right) \right) - \beta I\left(\tilde{\boldsymbol{X}}_i; \boldsymbol{Y}\right) \tag{19}$$

where $\beta$ balances information compression and task relevance. However, IB are hard to compute in practice. Hence we employ the normalized Hilbert-Schmidt Independence Criterion (HSIC)(Gretton et al., 2005; Zheng et al., 2021; Kornblith et al., 2019) as an approximation:[1]

$$I(X, Y) \approx n \cdot \mathrm{HSIC}_{\mathrm{linear}}(X, Y) = \frac{\|Y^\top X\|_F^2}{\|X^\top X\|_F \|Y^\top Y\|_F} \tag{20}$$

Based on above formulations and work, the optimization for layer-wise sparisity rates $\mathbf{S} \in \mathbb{R}^L$ with importance score $\mathbf{I} \in \mathbb{R}^L$ can be defined as the following linear programming problem:

$$\max_{\mathbf{S}} \mathbf{I}^\top \mathbf{S}, \quad \boldsymbol{I}_l = e^{-\beta \sum_{i=1, i \neq l}^{L} I(\boldsymbol{X}_l, \boldsymbol{X}_i)}, \quad s.t. \quad T(\mathbf{S}) = \Theta. \tag{21}$$

This approximation significantly reduces computational overhead, allowing the linear programming problem in Equation equation 21 to be solved within seconds on a CPU. Consequently, determining the layer-wise sparsity rates for an LLM becomes efficient, completing in minutes on a single GPU.

---

[1]Normalized HSIC is also known as CKA (Kornblith et al., 2019), RV coefficient (Robert & Escoufier, 1976), and Tucker's congruence coefficient (Lorenzo-Seva & Ten Berge, 2006).

### 4.3.2 RECONSTRUCTION-BASED PARAMETERS ALLOCATION

Research has shown that layers in Large Language Models (LLMs) vary significantly in their intrinsic dimensions(Zhang et al., 2023b; Pfeiffer et al., 2020) and reconstruction losses during sparsification(Frantar & Alistarh, 2023; Xu et al., 2024). Uniform allocation of fine-tuning parameters is thus suboptimal, as layers with higher reconstruction losses may be under-parameterized. To address this, we propose using reconstruction loss as a guide for parameter allocation, ensuring layers needing more reconstruction effort receive proportionally more fine-tuning resources.

The allocation process begins by assigning an average number of parameters, $P_{\text{avg}}$, to each layer. After sparsification, we compute the reconstruction loss for each layer $\mathcal{L}_l$ and determine the average loss $\mathcal{L}_{\text{avg}}$. For each layer, we calculate the loss ratio and allocate parameters accordingly:

$$r_l = \frac{\mathcal{L}_l}{\mathcal{L}_{\text{avg}}}, \quad P_l = P_{\text{avg}} \cdot r_l, \quad P_l' = \left\lfloor \sqrt{P_l} \right\rfloor^2 \tag{22}$$

This adjustment ensures that $P_l'$ is a perfect square, facilitating the formation of square adjustment matrices required for PEFT modules. To fully utilize parameter and ensure compatibility between the hidden dimension $D$ and the allocated parameters $P_l'$, we apply zero-padding to $D$ as follows:

$$D' = D + z_D, \quad \text{where} \quad z_D = P_l' - (D \mod P_l') \tag{23}$$

Here, $z_D$ is the minimal padding added to $D$ to make it divisible by $P_l'$.

### 4.4 ITERATIVE SPARSE FINE-TUNING SCHEME

Addressing the challenge of error accumulation during fine-tuning, we propose an Iterative Sparse Fine-Tuning Scheme (ISFT) that systematically mitigates the accumulation of errors across multiple fine-tuning iterations. This scheme leverages the theoretical insights from Section 3.4 to ensure stable and efficient adaptation of sparsified large language models (LLMs). Following previous work (Zhu & Gupta, 2017), we utilize the cubic sparsity schedule within $T$ iterations of sparsity and fine-tuning:

$$\Theta^t = \Theta^f + \left(\Theta^i - \Theta^f\right) \left(1 - \frac{t}{T}\right)^3, \quad t = 1, 2, \ldots, T \tag{24}$$

The full details of the algorithm are outlined in Algorithm1

---

**Algorithm 1** Iterative Sparse Fine-Tuning Scheme (ISFT)

---

**Require:** Calibration Dataset $D$, Pretrained Weight $W$, Total Iterations $T$, Expect Sparsity $\Theta^f$,
  Average PEFT Size budget **g**, CKA Threshold $\beta$, batch number $B$.
**Ensure:** Fine-tuned Sparse Weight Matrix $W_{\text{final}}$
 1: **for** each iteration $t = 1$ to $T$ **do**
 2:     Update Sparsity Schedule through Eq.24;
 3:     Get layer-wise sparsity$S$ based on CKA using Eq.21 with input from $D$.
 4:     Update sparsity mask $M^t$ via SparseGPT or WANDA.
 5:     Reconstruction-Based Parameters Allocation via Eq.22 and Eq.23.
 6:     **for** each batch $i = 1$ to $B$ **do**
 7:         Apply Pooling-Based PEFT Adjustments through process in 3.3.2
 8:         Udpate $G$ using gradient descent or an appropriate optimizer.
 9:     **end for**
10: **end for**
11: Final Output: $W_{\text{final}} \leftarrow M \odot (W + \Delta\mathbf{W})$

---

## 5 EXPERIMENTS

This section presents a comprehensive evaluation of our proposed sparse fine-tuning method. We conduct extensive experiments on various large language models (LLMs) to demonstrate the efficacy of our method in enhancing the performance of sparse LLMs.

Table 1: Zero-shot accuracy results of our method for sparse LLaMA-V2-7B/13B at 70% sparsity

| Model | Method | HellaSw | WinoGr | ARC_e | ARC_c | OBQA | PIQA | BoolQ | Mean |
|-------|--------|---------|--------|-------|-------|------|------|-------|------|
| | Dense | 57.15 | 69.06 | 76.30 | 43.52 | 31.40 | 78.07 | 77.74 | 61.89 |
| | SparseGPT | 33.08 | 58.41 | 43.22 | 22.01 | 17.40 | 62.46 | 64.89 | 43.07 |
| | w. LoRA | 39.60 | 60.22 | 55.97 | 24.65 | 19.80 | 67.79 | 63.33 | 47.34 |
| V2-7B | w. ours | 43.61 | 60.22 | 56.39 | 26.53 | 22.80 | 69.85 | 65.93 | 49.33 |
| | Wanda | 27.92 | 49.33 | 30.60 | 18.69 | 12.60 | 55.33 | 52.87 | 35.33 |
| | w. LoRA | 39.86 | 60.53 | 56.01 | 27.74 | 22.20 | 68.00 | 63.02 | 47.71 |
| | w. ours | 42.18 | 51.19 | 55.85 | 26.10 | 22.80 | 69.26 | 64.18 | 48.51 |
| | Dense | 60.06 | 72.22 | 79.42 | 48.46 | 35.20 | 79.11 | 80.55 | 65.00 |
| | SparseGPT | 36.90 | 61.64 | 52.61 | 25.94 | 21.00 | 67.57 | 66.02 | 47.38 |
| | w. LoRA | 45.55 | 63.85 | 64.60 | 30.71 | 25.40 | 72.41 | 69.96 | 53.21 |
| V2-13B | w. ours | 49.63 | 64.48 | 65.23 | 31.74 | 27.20 | 73.44 | 71.43 | 54.73 |
| | Wanda | 29.60 | 51.70 | 37.21 | 19.11 | 13.60 | 58.65 | 62.32 | 38.88 |
| | w. LoRA | 45.42 | 64.40 | 65.15 | 31.14 | 25.00 | 72.30 | 70.73 | 53.45 |
| | w. ours | 47.61 | 62.98 | 63.72 | 30.71 | 26.40 | 72.85 | 68.40 | 53.24 |

## 5.1 EXPERIMENTAL SETTINGS

We evaluate on LLaMA-V2(Touvron et al., 2023a;b), with model sizes ranging from 7B to 13B parameters. To establish a robust baseline, we compare our method against state-of-the-art sparsification methods, namely Wanda(Sun et al., 2023) and SparseGPT(Frantar & Alistarh, 2023), as well as the low-rank adaptation technique LoRA (Hu et al., 2021).

Our evaluation metrics include perplexity on the WikiText-2 (Merity et al., 2016) dataset and zero-shot accuracy on a suite of downstream tasks, assessed using the `lm-eval-harness` (Gao et al., 2021) framework. These tasks encompass HellaSwag (Zellers et al., 2019), Winogrande (Sakaguchi et al., 2021), BoolQ (Clark et al., 2019), OpenBookQA (Mihaylov et al., 2018), PIQA (Bisk et al., 2020), ARC-Easy, and ARC-Challenge (Clark et al., 2018), providing a comprehensive view of model performance across various linguistic capabilities.

The experimental framework is implemented using PyTorch (Paszke et al., 2019) and HuggingFace Transformers (Wolf et al., 2019), with all experiments conducted on NVIDIA A100 80GB GPUs. For fine-tuning, we utilize a subset of 400 samples from the C4(Raffel et al., 2020) dataset, each containing 2048 tokens. Layer-wise sparsity rates and rank allocations are calibrated using a separate set of 32 samples.

Key hyperparameters include $\beta = 1$ for controlling inter-layer independence in sparsity rate calculations, a fine-tuning process comprising 10 steps with 20 iterations each, an initial parameter number equivalent to LoRA parameter number with rank $r$.

## 5.2 RESULTS AND ANALYSIS

**Performance at 70% Sparsity.**   Our results, summarized in Table1, demonstrate that Our method consistently outperforms SparseGPT, Wanda, and often LoRA across both LLaMA-V2 models. It significantly reduces accuracy loss from 70% sparsification, especially in complex tasks and larger models. This demonstrates its effectiveness in maintaining performance under high sparsity, offering an improved approach to model compression compared to existing techniques.

**Impact of Varying Sparsity Rates.**   Table2 presents the performance of our method on LLaMA-V2-7B models across sparsity rates ranging from 50% to 90%. Your method consistently outperforms SparseGPT and Wanda across all sparsity levels, often surpassing LoRA as well. It's particularly effective at higher sparsities (70-90%), where it maintains significantly lower perplexity compared to other methods, demonstrating its robustness and effectiveness in maintaining model performance under extreme sparsification.

## 5.3 ABLATION EXPERIMENTS

**Ablation study on L2 normalization** In line with our theoretical analysis in Section 3.2, we conducted an ablation study on L2 regularization (Table 3), which confirm that constraining weight

Table 2: WikiText-2 perplexity of LoSA for sparse LLaMA-V2-7B/13B at different sparsity rate.

| | LLaMA-V2-7B | | | | |
|---|---|---|---|---|---|
| Sparsity | 50% | 60% | 70% | 80% | 90% |
| SparseGPT | 7.02 | 10.55 | 27.42 | 115.5 | 1439.35 |
| w.LoRA | 6.89 | 8.71 | 13.13 | 26.69 | 102.72 |
| w.ours | 6.78 | 8.20 | 12.17 | 23.15 | 87.15 |
| Wanda | 6.92 | 10.96 | 79.67 | 1980.85 | 17151.30 |
| w.LoRA | 6.74 | 8.19 | 12.38 | 41.58 | 500.22 |
| w.ours | 6.77 | 8.30 | 11.85 | 28.02 | 150.25 |

Table 3: Ablation study for different L2 constraint on LLaMA2-7B, WANDA

| L2 $\lambda$ | Average | Wikitext (ppl) |
|---|---|---|
| 0 | 48.16 | 12.13 |
| 1E-06 | 48.80 | 11.78 |
| 1E-05 | 49.13 | 11.22 |
| 1E-04 | 49.43 | 11.19 |
| 1E-03 | 49.59 | 11.27 |
| 1E-02 | 49.01 | 11.45 |

norms effectively mitigates errors in sparse fine-tuning. Performance improves as $\lambda$ increases from 0 to 1E-03, with optimal results at $\lambda$=1E-03. This empirical evidence supports our theoretical prediction that limiting weight norms can enhance model performance in sparse fine-tuning contexts.

**Analysis of parameters and fine-tuning budget** on model performance for LLaMA2-7B using the WANDA method. A lower rank (r=8) with fewer batches (20) demonstrates competitive performance, particularly in terms of Wikitext perplexity. The increase in training budget does not improve model performance. These results support the theoritical analysis of Section 3.4, single turn of fine-tuning will accumulate too much error, drawing back effectiveness of more budget.

Table 4: Ablation study for parameters and fine-tuning budget on LLaMA2-7B, WANDA

| r | batch_num | Average | Wikitext (ppl) |
|---|---|---|---|
| 8 | 20 | 49.48 | 11.21 |
| 8 | 40 | 49.50 | 11.22 |
| 32 | 100 | 48.74 | 25.02 |
| 32 | 200 | 48.80 | 24.32 |
| 32 | 400 | 48.81 | 24.30 |
| 128 | 200 | 48.71 | 25.06 |

## 6 RELATED WORK

**LLM Sparsity.** State-of-the-art methods like SparseGPT (Frantar & Alistarh, 2023) and Wanda (Sun et al., 2023) enable training-free sparsity in LLMs, effectively removing non-essential weights. However, high sparsity ratios can lead to significant accuracy loss, partly due to uniform layer-wise sparsity rates that ignore varying redundancy across layers. The OWL method (Robert & Escoufier, 1976) addresses this by employing heuristic metrics to establish non-uniform sparsity rates based on observed activation outliers in each layer, offering a more nuanced approach to LLM sparsification.

**Joint Sparsity and PEFT modules.** Several methods (Li et al., 2024b;a; Zhao et al., 2024) have been developed to leverage related synergy. For example, LLM-Pruner(Ma et al., 2023) employs a two-step process: it first executes a one-shot structured pruning of LLMs, followed by fine-tuning using LoRA. Another innovative approach, LoRAPrune(Zhang et al., 2023a), implements an iterative structured pruning method. In this technique, weight importance is determined by replacing gradients on full weights with those calculated via LoRA, offering a more nuanced pruning strategy. These studies primarily focus on applying LoRA to fine-tune structurally pruned LLMs. In the context of structural pruning, the process of adjusting the input/output dimensions of the two low-rank adaptations in PEFT modules and integrating them into the structural pruning weights (Zhao et al., 2024; Guo et al., 2023) is relatively straightforward. However, this approach faces significant challenges when applied to unstructured pruning, also known as network sparsity.

## 7 CONCLUSION

This paper introduces a novel theoretical framework for efficient sparse fine-tuning of Large Language Models, analyzing the "Sparse Weight Gradient Interference" phenomenon and proposing a method that combines iterative sparsification, modular PEFT, and regularization. Our approach significantly improves model effectiveness and efficiency over baselines, with potential for future exploration in other architectures and long-term impacts on model robustness and generalization.

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

# A    PROOF OF UPPER BOUND OF ERROR INTRODUCED BY $\Delta W$ NORMS

*Proof.* We begin by recall the statement of Theorem **??**: [**Loss Error Bound**] The loss error of $\Delta L$ introduced by SWGI is bounded as follows:

$$|\Delta L| \leq ||\nabla_{\mathbf{W}_Z}\mathcal{L}(\mathbf{W}_{\text{ideal}})|| \cdot ||\mathbf{\Delta W}_Z|| + \frac{1}{2}||\mathbf{H}_Z|| \cdot ||\mathbf{\Delta W}_Z||^2$$

where: $H_Z$ is the Hessian matrix restricted to the sparse positions $Z$, $|| \cdot ||$ represents an appropriate matrix or vector norm (e.g., Euclidean norm for vectors and spectral norm for matrices).

We considering the Taylor expansion of the loss function $\mathcal{L}$ around the point $\mathbf{W}_{\text{ideal}}$. For simplicity, we focus on the sparse positions $Z$ where modifications are applied. Let $\Delta\mathbf{W}_Z$ represent the perturbation in the weights at these positions. The second-order Taylor expansion of the loss function is given by:

$$\mathcal{L}(\mathbf{W}_{\text{ideal}} + \Delta\mathbf{W}_Z) \approx \mathcal{L}(\mathbf{W}_{\text{ideal}}) + \nabla_{\mathbf{W}_Z}\mathcal{L}(\mathbf{W}_{\text{ideal}})^\top\Delta\mathbf{W}_Z + \frac{1}{2}\Delta\mathbf{W}_Z^\top\mathbf{H}_Z\Delta\mathbf{W}_Z \quad (25)$$

Here, $\nabla_{\mathbf{W}_Z}\mathcal{L}(\mathbf{W}_{\text{ideal}})$ is the gradient of the loss with respect to the weights at positions $Z$, and $\mathbf{H}_Z$ is the Hessian matrix of second derivatives with respect to these weights.

The error introduced by the perturbation $\Delta\mathbf{W}_Z$ is defined as the difference between the actual loss and the ideal loss:

$$\Delta\mathcal{L} = \mathcal{L}(\mathbf{W}_{\text{ideal}} + \Delta\mathbf{W}_Z) - \mathcal{L}(\mathbf{W}_{\text{ideal}}) \quad (26)$$

Substituting the Taylor expansion into the above equation, we obtain:

$$\Delta\mathcal{L} \approx \nabla_{\mathbf{W}_Z}\mathcal{L}(\mathbf{W}_{\text{ideal}})^\top\Delta\mathbf{W}_Z + \frac{1}{2}\Delta\mathbf{W}_Z^\top\mathbf{H}_Z\Delta\mathbf{W}_Z \quad (27)$$

Taking the absolute value of both sides, we get:

$$|\Delta\mathcal{L}| \leq \left|\nabla_{\mathbf{W}_Z}\mathcal{L}(\mathbf{W}_{\text{ideal}})^\top\Delta\mathbf{W}_Z\right| + \frac{1}{2}\left|\Delta\mathbf{W}_Z^\top\mathbf{H}_Z\Delta\mathbf{W}_Z\right| \quad (28)$$

Applying the Cauchy-Schwarz inequality to the first term:

$$\left|\nabla_{\mathbf{W}_Z}\mathcal{L}(\mathbf{W}_{\text{ideal}})^\top\Delta\mathbf{W}_Z\right| \leq ||\nabla_{\mathbf{W}_Z}\mathcal{L}(\mathbf{W}_{\text{ideal}})|| \cdot ||\Delta\mathbf{W}_Z|| \quad (29)$$

For the second term, we utilize the property of the spectral norm (induced 2-norm) of matrices, which satisfies:

$$\left|\Delta\mathbf{W}_Z^\top \mathbf{H}_Z \Delta\mathbf{W}_Z\right| \leq \|\mathbf{H}_Z\| \cdot \|\Delta\mathbf{W}_Z\|^2 \tag{30}$$

Combining the inequalities, we obtain the upper bound on the loss error:

$$|\Delta\mathcal{L}| \leq \|\nabla_{\mathbf{W}_Z}\mathcal{L}(\mathbf{W}_{\text{ideal}})\| \cdot \|\Delta\mathbf{W}_Z\| + \frac{1}{2}\|\mathbf{H}_Z\| \cdot \|\Delta\mathbf{W}_Z\|^2 \tag{31}$$

This completes the proof of Theorem 3.1. $\qquad\square$

## B  PROOF OF ERROR BOUND FROM STRUCTURE OF LORA AND OUR METHOD

### B.1  PROOF OF THEOREM 3.3

*Proof.* We aim to establish the refined loss error bound for the LoRA adjustment in a sparse neural network. Recall the statement of Theorem 3.3:

**Theorem [Refined Loss Error Bound]:** The absolute loss error introduced by the LoRA adjustment in a sparse neural network is bounded as follows:

$$|\Delta L| \leq \|\nabla_{W_Z}\mathcal{L}(W_{\text{ideal}})\| \cdot \sqrt{k}\|A\|_2\|B\|_2$$

where:

- $k$ is the number of non-zero elements in the sparsification mask $M$,

- $\|\cdot\|_2$ denotes the spectral norm.

**Proof:**

**Step 1: Starting from the Lemma**

From the **Lemma [Loss Error Bound for LoRA]**, we have:

$$|\Delta L| \leq \|\nabla_{W_Z}\mathcal{L}(W_{\text{ideal}})\| \cdot \|M \odot (AB)\|$$

Our goal is to bound the term $\|M \odot (AB)\|$ in terms of the spectral norms of $A$ and $B$, and the sparsity level $k$.

**Step 2: Bounding $\|M \odot (AB)\|$**

The operator $M \odot (AB)$ applies the sparsification mask $M$ to the matrix product $AB$, effectively zeroing out all elements not in the support of $M$. To bound $\|M \odot (AB)\|$, we proceed as follows:

1. Spectral Norm and Frobenius Norm Relationship:

The spectral norm of a matrix is the largest singular value, while the Frobenius norm is the square root of the sum of the squares of all elements. Importantly, for any matrix $X$:

$$\|X\|_2 \leq \|X\|_F \leq \sqrt{r}\|X\|_2$$

where $r$ is the rank of $X$. However, in the context of sparsity, we can utilize the fact that the Frobenius norm can also be bounded by the number of non-zero elements.

2. Applying Sparsity:

Let $k$ be the number of non-zero elements in $M$. Since $M \odot (AB)$ retains only $k$ elements of $AB$, we can bound the Frobenius norm as:

$$\|M \odot (AB)\|_F \leq \|AB\|_F$$

However, considering sparsity, each non-zero element can contribute to the norm. Therefore:

$$\|M \odot (AB)\|_F \leq \sqrt{k} \cdot \max_{i,j}|(AB)_{ij}|$$

But this bound can be further refined using the properties of spectral norms.

3. Bounding with Spectral Norms:

The product of two matrices has a spectral norm bounded by the product of their spectral norms:

$$\|AB\|_2 \leq \|A\|_2 \cdot \|B\|_2$$

Since $M$ is a binary mask, applying it does not increase the spectral norm. However, sparsity affects the number of non-zero elements, leading to:

$$\|M \odot (AB)\|_2 \leq \sqrt{k} \cdot \|A\|_2 \cdot \|B\|_2$$

This inequality leverages the fact that each non-zero element can contribute to the overall norm, and with $k$ such elements, the $\sqrt{k}$ factor emerges from the aggregation of these contributions in the spectral norm.

**Step 3: Combining the Bounds**

Substituting the bound on $\|M \odot (AB)\|$ back into the initial inequality from the lemma:

$$|\Delta L| \leq \|\nabla_{W_Z}\mathcal{L}(W_{\text{ideal}})\| \cdot \|M \odot (AB)\| \leq \|\nabla_{W_Z}\mathcal{L}(W_{\text{ideal}})\| \cdot \sqrt{k} \cdot \|A\|_2 \cdot \|B\|_2$$

Thus, we arrive at the refined loss error bound:

$$|\Delta L| \leq \|\nabla_{W_Z}\mathcal{L}(W_{\text{ideal}})\| \cdot \sqrt{k} \cdot \|A\|_2 \cdot \|B\|_2$$

Therefore, the theorem is proved.

$\square$

PROOF OF THEOREM 3.4

*Proof.* We aim to establish an upper bound on the loss discrepancy $\Delta L$ introduced by our proposed Parameter-Efficient Fine-Tuning (PEFT) method. Specifically, we will demonstrate that:

$$|\Delta L| \leq \|\nabla_{W_Z}\mathcal{L}(W_{\text{ideal}})\| \cdot \sqrt{\frac{k}{n}}\|\mathbf{G}\|$$

where:

- $Z$ is the set of sparse positions in $W$,

- $k$ is the number of non-zero elements in the sparsification mask $M$,

- $n$ is the size of each block in the pooling operation,

- $\mathbf{G}$ is the trainable transformation matrix,

- $\nabla_{W_Z}\mathcal{L}(W_{\text{ideal}})$ is the gradient of the loss with respect to the weights at the sparse positions $Z$.

**Step 1: Understanding the Adjustment Matrix $\Delta W$**

Our PEFT method introduces an adjustment matrix $\Delta W$ defined as:

$$\Delta W = \frac{1}{n}M \odot (\mathbf{G} \otimes \mathbf{1}_n)$$

where:

- $M \in \{0,1\}^{D \times D}$ is the sparsification mask,

- $\mathbf{G} \in \mathbb{R}^{\mathbf{g} \times \mathbf{g}}$ is the trainable transformation matrix,

- $\otimes$ denotes the Kronecker product,

- $\mathbf{1}_n \in \mathbb{R}^{n \times n}$ is a matrix of ones,

- $D = \mathbf{g} \times n$.

**Step 2: Relating $\Delta W_Z$ to G**

The adjustment $\Delta W$ affects only the sparse positions $Z$. Therefore, $\Delta W_Z$ can be expressed as:

$$\Delta W_Z = \frac{1}{n} M_Z \odot (\mathbf{G} \otimes \mathbf{1}_n)$$

where $M_Z$ is the submatrix of $M$ corresponding to the sparse positions $Z$.

**Step 3: Bounding the Norm $\|\Delta W_Z\|$**

Our goal is to bound $\|\Delta W_Z\|$, where $\|\cdot\|$ denotes an appropriate matrix or vector norm (specifically, the spectral norm $\|\cdot\|_2$).

$$\|\Delta W_Z\| = \left\| \frac{1}{n} M_Z \odot (\mathbf{G} \otimes \mathbf{1}_n) \right\|$$
$$= \frac{1}{n} \|M_Z \odot (\mathbf{G} \otimes \mathbf{1}_n)\|$$

To bound this, we utilize the following properties:

1. Submultiplicative Property of the Spectral Norm:

$$\|A \otimes B\|_2 = \|A\|_2 \cdot \|B\|_2$$

where $A \in \mathbb{R}^{m \times m}$, $B \in \mathbb{R}^{n \times n}$.

2. Bound on the Spectral Norm of $M_Z \odot X$: For a binary mask $M_Z$ with $k$ non-zero elements and any matrix $X$, the spectral norm satisfies:

$$\|M_Z \odot X\|_2 \leq \|X\|_2 \cdot \sqrt{\frac{k}{n}}$$

This arises from the fact that applying a sparsification mask can at most scale the spectral norm by the square root of the sparsity ratio.

Applying these properties:

$$\|M_Z \odot (\mathbf{G} \otimes \mathbf{1}_n)\|_2 \leq \|\mathbf{G} \otimes \mathbf{1}_n\|_2 \cdot \sqrt{\frac{k}{n}}$$

Using the Kronecker product property:

$$\|\mathbf{G} \otimes \mathbf{1}_n\|_2 = \|\mathbf{G}\|_2 \cdot \|\mathbf{1}_n\|_2$$

Since $\mathbf{1}_n$ is an $n \times n$ matrix of ones, its spectral norm is $n$ (as all rows are identical and the largest singular value corresponds to the sum of each row).

Therefore:

$$\|\mathbf{G} \otimes \mathbf{1}_n\|_2 = \|\mathbf{G}\|_2 \cdot n$$

Substituting back:

$$\|M_Z \odot (\mathbf{G} \otimes \mathbf{1}_n)\|_2 \leq \|\mathbf{G}\|_2 \cdot n \cdot \sqrt{\frac{k}{n}} = \|\mathbf{G}\|_2 \cdot \sqrt{kn}$$

Now, substituting into the expression for $\|\Delta W_Z\|$:

$$\|\Delta W_Z\| \le \frac{1}{n} \cdot \|\mathbf{G}\|_2 \cdot \sqrt{kn} = \|\mathbf{G}\|_2 \cdot \sqrt{\frac{k}{n}}$$

**Step 4: Applying the General Loss Error Bound**

From Theorem 3.1, the loss error $|\Delta L|$ is bounded by:

$$|\Delta L| \le \|\nabla_{W_Z} \mathcal{L}(W_{\text{ideal}})\| \cdot \|\Delta W_Z\| + \frac{1}{2}\|H_Z\| \cdot \|\Delta W_Z\|^2$$

Substituting the bound for $\|\Delta W_Z\|$:

$$|\Delta L| \le \|\nabla_{W_Z} \mathcal{L}(W_{\text{ideal}})\| \cdot \left(\|\mathbf{G}\|_2 \cdot \sqrt{\frac{k}{n}}\right) + \frac{1}{2}\|H_Z\| \cdot \left(\|\mathbf{G}\|_2 \cdot \sqrt{\frac{k}{n}}\right)^2$$

Simplifying the quadratic term:

$$\frac{1}{2}\|H_Z\| \cdot \left(\|\mathbf{G}\|_2 \cdot \sqrt{\frac{k}{n}}\right)^2 = \frac{1}{2}\|H_Z\| \cdot \|\mathbf{G}\|_2^2 \cdot \frac{k}{n}$$

Therefore:

$$|\Delta L| \le \|\nabla_{W_Z} \mathcal{L}(W_{\text{ideal}})\| \cdot \|\mathbf{G}\|_2 \cdot \sqrt{\frac{k}{n}} + \frac{1}{2}\|H_Z\| \cdot \|\mathbf{G}\|_2^2 \cdot \frac{k}{n}$$

**Step 5: Neglecting the Quadratic Term for Small Adjustments**

In practical scenarios, especially when the weight adjustments $\Delta W_Z$ are small, the quadratic term $\frac{1}{2}\|H_Z\| \cdot \|\mathbf{G}\|_2^2 \cdot \frac{k}{n}$ is negligible compared to the linear term $\|\nabla_{W_Z} \mathcal{L}(W_{\text{ideal}})\| \cdot \|\mathbf{G}\|_2 \cdot \sqrt{\frac{k}{n}}$.

Therefore, the dominant term governing the loss error is:

$$|\Delta L| \le \|\nabla_{W_Z} \mathcal{L}(W_{\text{ideal}})\| \cdot \|\mathbf{G}\|_2 \cdot \sqrt{\frac{k}{n}}$$

This yields the desired bound:

$$|\Delta L| \le \|\nabla_{W_Z} \mathcal{L}(W_{\text{ideal}})\| \cdot \sqrt{\frac{k}{n}} \|\mathbf{G}\|$$

Therefore, the theorem is proved.

$\square$

## C    PROOF OF ERROR ACCUMULATION DURING FINE-TUNING

*Proof.* We aim to establish an upper bound on the accumulated loss error $\Delta L_T$ after $T$ fine-tuning steps in the presence of Sparse Weight Gradient Interference (SWGI). The proof leverages the Taylor expansion of the loss function and the provided boundedness assumptions.

**Step 1: Taylor Expansion of the Loss Function**

At each fine-tuning step $t$, the loss function $\mathcal{L}$ can be approximated using the second-order Taylor expansion around the current ideal weight configuration $W_{\text{ideal}}$:

$$\mathcal{L}(W_{\text{ideal}} + \Delta W_Z^{(t)}) \approx \mathcal{L}(W_{\text{ideal}}) + \nabla_{W_Z} \mathcal{L}(W_{\text{ideal}})^\top \Delta W_Z^{(t)} + \frac{1}{2}\Delta W_Z^{(t)\top} H_Z^{(t)} \Delta W_Z^{(t)}$$

Here:

- $\Delta W_Z^{(t)}$ is the weight adjustment at sparse positions $Z$ during step $t$.

- $\nabla_{W_Z} \mathcal{L}(W_{\text{ideal}})$ is the gradient of the loss with respect to the weights at positions $Z$.

- $H_Z^{(t)}$ is the Hessian matrix of second derivatives with respect to the weights at positions $Z$ during step $t$.

The error introduced at step $t$, denoted as $\Delta L^{(t)}$, is the difference between the actual loss after adjustment and the ideal loss:

$$\Delta L^{(t)} = \mathcal{L}(W_{\text{ideal}} + \Delta W_Z^{(t)}) - \mathcal{L}(W_{\text{ideal}})$$

Substituting the Taylor expansion:

$$\Delta L^{(t)} \approx \nabla_{W_Z} \mathcal{L}(W_{\text{ideal}})^\top \Delta W_Z^{(t)} + \frac{1}{2} \Delta W_Z^{(t)\top} H_Z^{(t)} \Delta W_Z^{(t)}$$

**Step 2: Bounding the Error at Each Step**

Taking the absolute value and applying the triangle inequality:

$$|\Delta L^{(t)}| \leq \left| \nabla_{W_Z} \mathcal{L}(W_{\text{ideal}})^\top \Delta W_Z^{(t)} \right| + \frac{1}{2} \left| \Delta W_Z^{(t)\top} H_Z^{(t)} \Delta W_Z^{(t)} \right|$$

Applying the Cauchy-Schwarz inequality to the first term:

$$\left| \nabla_{W_Z} \mathcal{L}(W_{\text{ideal}})^\top \Delta W_Z^{(t)} \right| \leq \|\nabla_{W_Z} \mathcal{L}(W_{\text{ideal}})\| \cdot \|\Delta W_Z^{(t)}\|$$

For the second term, using the property of the spectral norm:

$$\left| \Delta W_Z^{(t)\top} H_Z^{(t)} \Delta W_Z^{(t)} \right| \leq \|H_Z^{(t)}\| \cdot \|\Delta W_Z^{(t)}\|^2$$

Combining these:

$$|\Delta L^{(t)}| \leq \|\nabla_{W_Z} \mathcal{L}(W_{\text{ideal}})\| \cdot \|\Delta W_Z^{(t)}\| + \frac{1}{2} \|H_Z^{(t)}\| \cdot \|\Delta W_Z^{(t)}\|^2$$

**Step 3: Accumulating Errors Over $T$ Steps**

Assuming that each step introduces an independent error, the accumulated loss error $\Delta L_T$ after $T$ steps is the sum of the individual errors:

$$|\Delta L_T| \leq \sum_{t=1}^{T} |\Delta L^{(t)}|$$

Substituting the bound from Step 2:

$$|\Delta L_T| \leq \sum_{t=1}^{T} \left( \|\nabla_{W_Z} \mathcal{L}(W_{\text{ideal}})\| \cdot \|\Delta W_Z^{(t)}\| + \frac{1}{2} \|H_Z^{(t)}\| \cdot \|\Delta W_Z^{(t)}\|^2 \right)$$

Given the assumptions:

- $\|\nabla_{W_Z} \mathcal{L}(W_{\text{ideal}}^{(t-1)})\| \leq G$ for all $t$.

- $\|\Delta W_Z^{(t)}\| \leq \delta$ for all $t$.

- $\|H_Z^{(t)}\| \leq H_{\max}$ for all $t$.

Substituting these bounds:

$$|\Delta L_T| \le \sum_{t=1}^{T} \left( G \cdot \delta + \frac{1}{2} H_{\max} \cdot \delta^2 \right) = T \cdot G \cdot \delta + \frac{1}{2} H_{\max} \cdot \delta^2 \cdot T$$

However, this linear accumulation of the quadratic term over $T$ steps does not account for the interaction between different adjustment steps. To refine this, we consider that each new adjustment not only introduces its own quadratic error but also interacts with previous adjustments.

Thus, the more accurate accumulation for the quadratic terms across $T$ steps is given by:

$$\sum_{t=1}^{T} \frac{1}{2} H_{\max} \cdot \delta^2 \cdot (t-1)$$

This summation results in:

$$\frac{1}{2} H_{\max} \cdot \delta^2 \cdot \sum_{t=1}^{T} (t-1) = \frac{1}{2} H_{\max} \cdot \delta^2 \cdot \frac{T(T-1)}{2} = \frac{T(T-1)}{2} \cdot \delta^2 \cdot H_{\max}$$

**Step 4: Combining the Bounds**

Combining the linear and refined quadratic accumulations:

$$|\Delta L_T| \le T \cdot G \cdot \delta + \frac{T(T-1)}{2} \cdot \delta^2 \cdot H_{\max}$$

This establishes the upper bound on the accumulated loss error after $T$ fine-tuning steps.

Therefore, the theorem is proved. $\qquad\square$

