# OpenReview forum: "Mitigating Gradient Interference for Efficient Sparse Fine-Tuning of Large Language Models"
_ICLR.cc/2025/Conference — Submitted to ICLR 2025_

### Official Review · Reviewer_FRnz · 2024-10-28

**Soundness:** 2
**Presentation:** 2
**Contribution:** 2
**Rating:** 5
**Confidence:** 4

**Summary:**

This paper provides a theoretical examination of parameter-efficient fine-tuning (PEFT) for sparse large language models (LLMs). The authors explore Sparse Weight Gradient Interference, identifying large weight norm as a potential main source of loss error in general sparse PEFT methods. Additionally, they suggest that the LoRA structure contributes to loss error in sparse PEFT and propose an alternative PEFT approach consisting of three steps—pooling, linear transformation, and expansion—which they argue achieves a tighter upper bound on loss error compared to LoRA. They also raise the possibility of error accumulation over fine-tuning iterations as a further source of loss. Alongside their theoretical insights, the authors present a brief empirical analysis showing that their sparse PEFT method can outperform LoRA across certain benchmarks when combined with the sparseGPT and Wanda LLM pruning techniques.

**Strengths:**

- The theoretical examination of Sparse Weight Gradient Interference in sparse PEFT methods is sound and provides valuable insights.
- The proposed method outperforms LoRA in specific method-benchmark combinations.
- The improvements apply across different levels of sparsity and sizes of language models.

**Weaknesses:**

- The paper lacks detailed experimental settings, making it unclear if the comparison with the baseline is entirely fair. Key details such as the number of data points used for LoRA, the number of fine-tuning iterations, hyper-parameters, and any tuning performed (particularly for learning rate) are missing.
- The empirical evaluation is limited to classification tasks, and additional open-ended generation downstream tasks would strengthen the assessment of the proposed method.
- Even on the limited set of reported downstream tasks, improvements are inconsistent, with LoRA outperforming the proposed method on certain model-task combinations, such as Llama2(13b)-wanda.
- There is no discussion on result variance; only single values are reported. Given the minor improvements observed, additional experiments with varying random seeds are needed to allow readers to assess the method's efficacy more reliably.
- The statement 'Empirical evaluation on a 50% sparse LLaMA-2 7B model demonstrates the superiority of our approach, achieving lossless compression' in abstract is misleading and not supported by the results.

**Questions:**

Please see the weaknesses section.

---

### Official Review · Reviewer_JXRk · 2024-11-04

**Soundness:** 3
**Presentation:** 2
**Contribution:** 2
**Rating:** 5
**Confidence:** 4

**Summary:**

In this paper, the authors focus on the sparsification and PEFT recovery of LLMs. The authors identify and address the Sparse Weight Gradient Interference (SWGI) phenomenon, where gradients from masked weights interfere with the fine-tuning of active parameters, leading to performance degradation. The authors conduct a theoretical analysis on this problem, and propose a new new iterative sparse fine-tuning scheme to handle this problem. Experiments on benchmarks show that the proposed method recovers the accuracy better than LoRA.

**Strengths:**

- The authors point out the Sparse Weight Gradient Interference (SWGI) phenomenon, attributing to the mismatch between the sparse structure of the model weights and the dense structure of the PEFT weights.

- The authors provide a theoretical analysis of the SWGI phenomenon and related bounds on the errors introduced by sparsification.

**Weaknesses:**

- The structure and readability of the paper could be improved. Also, a main figure illustrating the problem setting or the SWGI phenomenon, and the proposed method could be helpful.

- A detailed ablation study on the many components that comprise the method (CKA-guided sparsity setting, PEFT parameter allocation, sparsity scheduling) is missing.

- More experiments on well-accepted benchmarks such as MMLU or GSM8K are needed to verify the effectiveness of the proposed method.

**Questions:**

- How does the proposed method perform for structured or semi-structured sparsity?

---

### Official Review · Reviewer_eJLj · 2024-11-08

**Soundness:** 3
**Presentation:** 3
**Contribution:** 3
**Rating:** 5
**Confidence:** 4

**Summary:**

A common method to prune iteratively is to learn the mask and also update the model weights to recover the loss induced by pruning using fine-tuning. The mask is applied such that the reconstruction loss of each layer after applying the mask is minimized. Rather than regular fine-tuning, this method employs LoRA to heal the model. They posit that using LoRA to heal the model introduces errors owing to the "Sparse Weight Gradient Inference" problem. This occurs because the LoRA module is not aware of where the masks have been applied to the frozen pre-trained model. Thus, it could have gradients for parameters where the frozen pre-trained model is pruned and set to 0 resulting in interference. They estimate the error that using LoRA incurs.

The proposed method uses bilevel optimization where in the upper level the mask is learnt and in the lower level the weights are updated using a modified LoRA. The modified LoRA applies a pooling operation on the input to reduce it to a lower dimension g, followed by multiplying the resulting value with weight G and then projects the output back to the original dimension. The weights G are learnt in this PEFT variant. To further alleviate the error introduced by using LoRA to heal is to rein in the magnitude of weight change by using normalization such as weight-decay or drop out etc. Rather than introducing the parameters uniformly to all layers for fine-tuning, more parameters are allocated to layers with higher reconstruction loss. The layerwise sparsity rate is set based on the Hilbert-Schmidt Independence Criterion metric and is inversely proportional to it, thereby pruning layers with higher information redundancy. This whole process is repeated for several iterations and the layer-wise sparsity rate, the sparsity mask and model weights are updated at each iteration.

**Strengths:**

1. Allocating additional fine-tuning parameters to the layers with higher reconstruction loss is novel.
2. They demonstrate that the zero-shot performance of the model after pruning is the best when compared to the other baselines on various tasks.

**Weaknesses:**

1. While the sparsity is induced by applying masks, unstructured pruning does not reduce the latency. In real world applications, reducing the latency is also crucial. Could you report the latency improvements of the final model and how it compares to just using N:M sparsity and WANDA etc?

**Questions:**

1. Wanda does not require updating the model weights during pruning unlike the proposed method and SparseGPT. So if Wanda is used as a baseline here, is it also fine-tuned using the same amount of data as this method for fair comparison?
2. It is not clear in line 4 of algorithm 1 why the sparsity mask is updated using either SparseGPT or WANDA
3. Given that this method uses PEFT and does not have to update all the model parameters like SparseGPT, can you report the amount of time and memory required to run your method and compare it against other baselines too? This might bolster your case further.
4. Could you provide some insights on why is your method so much better than SparseGPT given that the latter updates the weights of the entire model?

---

### Official Review · Reviewer_3qDY · 2024-11-08

**Soundness:** 3
**Presentation:** 4
**Contribution:** 2
**Rating:** 3
**Confidence:** 3

**Summary:**

This paper introduce a comprehensive theoretical framework for memory-efficient fine-tuning for sparse LLMs. The authors identify and analyze a key challenge called "Sparse Weight Gradient Interference," where masked pre-trained weights and PEFT weights exhibit competing optimization objectives during fine-tuning. To address this, they propose a novel method combining three key innovations: a pooling-based PEFT method, normalization of PEFT modules, and an adaptive layer-wise approach using Centered Kernel Alignment for sparsity allocation. Their theoretical analysis identifies three crucial factors affecting fine-tuning efficacy: errors from weight norms, PEFT structures, and error accumulation during fine-tuning. The effectiveness of their approach is demonstrated through extensive experiments on LLaMA-2 models, showing superior performance compared to existing methods, particularly in maintaining model performance under high sparsity conditions (up to 70%), while providing theoretical guarantees for error bounds.

**Strengths:**

1. This paper have a strong theoretical foundation with mathematical proofs about the error bounds of using dense adapter for sparse LLMs. It emphasize on the gradient interference, and mitigate this error for a better PEFT method for sparse models.

2. Novel identification and analysis of the gradient interference problem. This also results in a novel PEFT method.

3. Experimental results show its advantage in comparison to LoRA.

4. This paper is well-written.

**Weaknesses:**

1. Lack empirical upper bound: The paper lacks a crucial empirical upper bound comparison. Since the core problem stems from using dense adapter weights with a sparse model, an ideal oracle baseline would be directly fine-tuning only the sparse positions in the model. Including this oracle baseline would provide a clearer understanding of the maximum achievable performance without gradient interference. This comparison would also serve as a valuable reference point for future research in sparse fine-tuning methods.

2. Limited model diversity in experiments: This work only evaluate LLaMA-2 family model. I will recommend the authors to further evaluate LLaMA-3 family and Mistral family for a more comprehensive comparison.

3. Lack practical efficiency: While the proposed method show better performance, the authors do not provide any efficiency results to show the practical training speed of the proposed method. The proposed method cannot be useful if it is slow even with strong theoretical foundation. Therefore, I think the authors should provide the number of trainable parameters, the training time, the training memory for LoRA and the proposed method.

4. Typo in Table 2: I don't see any other "LoSA" in this paper. The authors should either delete it or add an definition for it.

5. Lack of ablation studies: This paper do not do ablation study for the  Iterative Sparse Fine-Tuning Scheme. This should be added to verify its effectiveness.

I will consider to raise my score if more evidence are provided about the effectiveness of the proposed method.

**Questions:**

See weaknesses.

---

### Meta-Review · Area_Chair_32pE · 2024-12-18

**Metareview:**

This paper addresses the challenge of efficient fine-tuning for sparse large language models (LLMs) by introducing a novel theoretical framework. The authors identify and analyze the issue of Sparse Weight Gradient Interference (SWGI), which complicates the optimization of model weights during the fine-tuning process. This work aims to fill a critical gap in existing research and proposes a new iterative fine-tuning scheme to enhance model performance without substantial resource requirements.

The reviews provided a generally negative assessment, highlighting weaknesses in the paper's clarity, empirical validation, and lack of comprehensive comparisons with existing methods. Reviewers expressed differing views on the theoretical soundness and contributions, but all agreed that the empirical results were unsatisfactory. Notably, the reviewers pointed to the absence of an ablation study, insufficient experiments across diverse models, and unclear presentation of results.

Since the authors abandoned their rebuttal opportunities, many issues were left unresolved, leading all reviews to conclude that the paper fails to meet the acceptance criteria. Therefore, I recommend a rejection of this submission.

**Additional Comments On Reviewer Discussion:**

Since the authors abandoned their rebuttal opportunities, many issues were left unresolved, leading all reviews to conclude that the paper fails to meet the acceptance criteria. Therefore, I recommend a rejection of this submission.

---

### Decision · Program_Chairs · 2025-01-22

Reject